

# Who are the important predators of sea turtle nests at Wreck Rock beach?

Juan Lei and David T. Booth

School of Biological Sciences, The University of Queensland, Brisbane, St. Lucia, Australia

## ABSTRACT

Excessive sea turtle nest predation is a problem for conservation management of sea turtle populations. This study assessed predation on nests of the endangered loggerhead sea turtle (*Caretta caretta*) at Wreck Rock beach adjacent to Deepwater National Park in Southeast Queensland, Australia after a control program for feral foxes was instigated. The presence of predators on the nesting dune was evaluated by tracking plots (2 × 1 m) every 100 m along the dune front. There were 21 (2014–2015) and 41 (2015–2016) plots established along the dune, and these were monitored for predator tracks daily over three consecutive months in both nesting seasons. Predator activities at nests were also recorded by the presence of tracks on top of nests until hatchlings emerged. In addition, camera traps were set to record the predator activity around selected nests. The tracks of the fox (*Vulpes vulpes*) and goanna (*Varanus spp*) were found on tracking plots. Tracking plots, nest tracks and camera traps indicated goanna abundance varied strongly between years. Goannas were widely distributed along the beach and had a Passive Activity Index (PAI) (0.31 in 2014–2015 and 0.16 in 2015–2016) approximately seven times higher than that of foxes (PAI 0.04 in 2014–2015 and 0.02 in 2015–2016). Five hundred and twenty goanna nest visitation events were recorded by tracks but no fox tracks were found at turtle nests. Camera trap data indicated that yellow-spotted goannas (*Varanus panoptes*) appeared at loggerhead turtle nests more frequently than lace monitors (*V. varius*) did, and further that lace monitors only predated nests previously opened by yellow-spotted goannas. No foxes were recorded at nests with camera traps. This study suggests that large male yellow-spotted goannas are the major predator of sea turtle nests at the Wreck Rock beach nesting aggregation and that goanna activity varies between years.

## INTRODUCTION

Sea turtles are oviparous and construct their nests on dunes adjacent to the beach where embryos take about two month to incubate. Sea turtle hatchling nest emergence success is determined by nest temperature, salinity, humidity, water inundation and predation (*Fowler, 1979*; *Miller, 1985*; *Reid, Margaritoulis & Speakman, 2009*). During incubation, a wide range of predators may attack sea turtle nests and have a significant effect on hatchling recruitment and thus long-term population persistence (*Stancyk, 1995*). At many beaches, nest predation is the main cause of hatch failure of sea turtles with some regions reporting

Corresponding author
Juan Lei, lj881204@gmail.com

more than 50% of nests being destroyed by predators (e.g., *Fowler, 1979*; *Blamires & Guinea, 1998*; *Blamires, Guinea & Prince, 2003*; *Maulany, 2012*; *McLachlan et al., 2015*). A large variety of non-human species have been reported as sea turtle nest predators including, fire ants (*Solenopsis invicta*), crabs (*Ocypode cursor*), turkey vultures (*Cathartes aura*), black vultures (*Coragyps atratus*), coatis (*Nasua narica*), raccoons (*Procyon lotor*), dogs(*Canis familaris*), red foxes (*Vulpes vulpes*), golden jackals (*Canis aureus*), mongooses (*Herpestes javanicus*), snakes (*Oligodon formosanus*) and goannas (*Varanus spp*) in different regions of the world (*Stancyk, Talbert & Dean, 1980*; *Mora & Robinson, 1984*; *Brown & Macdonald, 1995*; *Frick, 2003*; *Leighton et al., 2008*). In Australia, sea turtle nest predators include several species of native goanna, the native dingo (*Canis lupus*) and the introduced fox (*Vulpes vulpes*), pig (*Sus scrofa*) and wild dog (*Canis familaris*) (*Limpus, 1978*; *Limpus & Fleay, 1983*). In particular, fox predation of sea turtle nests along the east Australian coast has been problematic and therefore a major focus of sea turtle conservation programs (*Limpus, 1978*; *Limpus & Fleay, 1983*; *Limpus, 2008*).

The loggerhead turtle (*Caretta caretta*) is an endangered species on the IUCN Red List (*IUCN, 2016*). Major breeding aggregations of loggerhead sea turtle include Africa-Mozambique, Oman, the Mediterranean sea, Sri Lanka, Japan, USA and Australia (*Limpus & Limpus, 2003*). Genetic studies indicate there is little or no interbreeding between these major breeding aggregations (*Bowen et al., 1993*; *Limpus, 2008*). In Australia, two genetically distinct breeding stocks have been identified: an eastern Australian population and western Australian population (*Limpus & Limpus, 2003*). If one breeding stock becomes extinct, it would be difficult to repopulate this area from other genetic stocks. In order to preserve the genetic diversity of loggerheads, it is necessary to protect each of the different populations.

A significant number of loggerhead turtles nest at Wreck Rock beach adjacent to Deepwater National Park, Queensland, Australia (~400 nests per season, *Limpus, 2008*). Predators of sea turtle nests at Wreck Rock beach include foxes, dingoes and goannas (*Limpus, 2008*). The fox predation of loggerhead turtle nests continued to increase from a modest level when monitoring commenced in 1968–1969 to 90–95% in the mid-1970s (*Limpus, 2008*). From 1987 onwards, 1,080 poison baits have been used to control fox predation (*Limpus, 2008*), but a recent nest survey (*McLachlan et al., 2015*) indicated that while fox predation of nests was minimal, a large number of nests were predated by goannas. The lace monitor (*Varanus varius*) and yellow-spotted goanna (*Varanus panoptes*) are likely to be the main goannas attacking loggerhead nests because of their distribution along the coastline and ability to dig holes while foraging (*Cogger, 1993*). However, the relative activity levels and impact of these species on loggerhead turtle nests at Wreck Rock beach remain unknown.

For some animal species, it is difficult to estimate population density by standard census methods such a mark and recapture (*Engeman & Allen, 2000*) because of large home ranges, rough terrain habitats, relatively sparse populations and/or difficulty in capturing animals or making direct observations (*Pelton & Marcum, 1977*). To overcome these problems, *Engeman & Allen (2000)* developed and refined a passive activity index (PAI) based on the occurrence of tracks on small, pre-defined plots of substrate for monitoring wild carnivore

species. This method is simple and quickly applied in the field and can also provide accurate information reflecting population changes over time or space, and simultaneously capture a suite of wildlife species (*Engeman & Allen, 2000*). This method has been used previously to monitor predator activities, including the common water monitor (*Varanus salvator*) activity on an olive ridley turtle (*Lepidochelys olivacea*) nesting beach in Indonesia over two nesting seasons (*Maulany, 2012*).

Despite the anecdotal evidence that foxes and more recently goannas predate a significant number of sea turtle nests at Wreck Rock beach (*Limpus, 2008*; *McLachlan et al., 2015*), no quantitative study of sea turtle nest predation has been conducted at this important nesting beach, and it is not known what species of goanna is responsible for predation. Therefore, the aim of this study was to fill this knowledge gap by quantifying goanna and fox activity on nesting dunes during the sea turtle nesting season at Wreck Rock beach. Three methods were used to achieve this aim. Firstly, tracking plots were used to monitor general activity levels of goannas and foxes along the dunes where sea turtles construct their nests. Secondly, turtle nests were inspected every day until turtle hatchlings emerged in order to record the activities of predators at the nest. Thirdly, camera traps were used to capture predator activity at sea turtle nests so that we could identify which species of goanna was the main predator of these nests.

## METHODS

### Study site and nest marking

This study was conducted along the beach for 3 km immediately to the north and south of Wreck Rock adjacent to Deepwater National Park, Southeast Queensland (24°18′58S, 151°57′55′E) (Fig. 1). This section of the beach is marked by numbered stakes every 100 m for ease of marking and relocating nests. The beach was monitored nightly by personnel from Turtle Care Volunteers Queensland Inc. to record the presence of emerging female turtles and successful nesting activities. When a nest was located, its position was marked by a red ribbon attached to a small stake and recorded using a handheld GPS (Garmin eTrex 30; Kansas, USA). All work was approved by a University of Queensland Animal Ethics Committee (permit #SBS/352/EHP/URG) and conducted under Queensland Government National parks scientific permit # WITK15315614.

### Tracking plots

Tracking plots were used to estimate relative activity of predators during the peak sea turtle nesting time across two consecutive years (5-Dec-2014 to 4-Mar-2015 and 30-Nov-2015 to 28-Feb-2016). In 2015–2016, these plots were also monitored for four days in April, a time when most sea turtle clutches had finished hatching. Twenty-one tracking plots (2 m × 1 m) in 2014–2015 and 41 in 2015–2016, spaced 100 m apart, were set up on the primary dune (where most sea turtle nests were constructed). The plots extended along the dunes for 1 km (2014–2015) and 2 km (2015–2016) north and south of Wreck Rock camping area. The monitored area of a plot was marked by sticks placed at each corner of the plot and the plot's location was recorded with a handheld GPS. Each plot was inspected daily during the afternoon (weather permitting), and the number of goanna and fox tracks

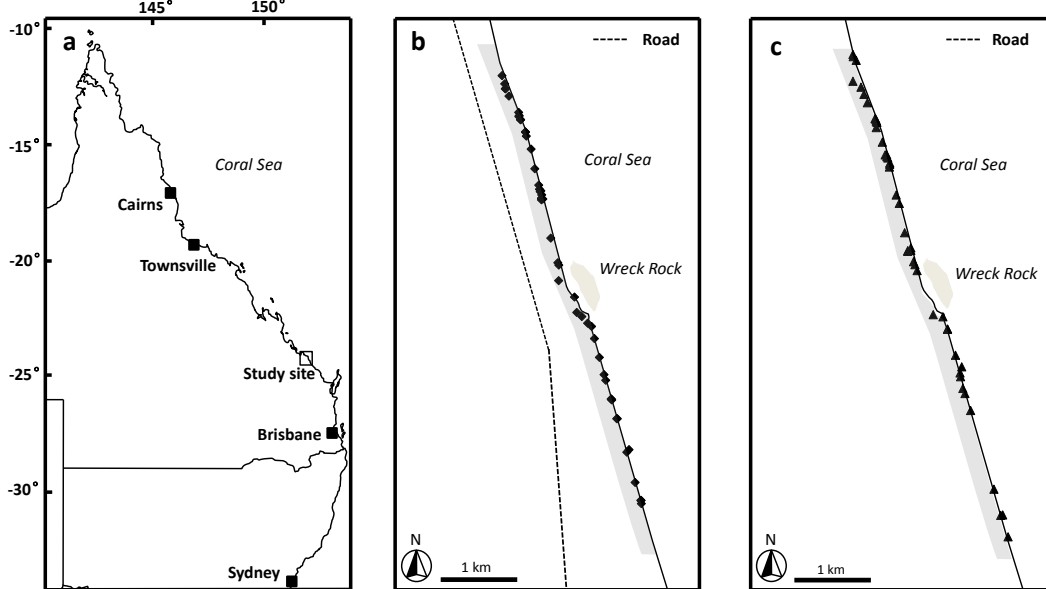

**Figure 1** **Image of study area.** (A) Location of study site, Wreck Rock beach adjacent to Deepwater National Park, Queensland, Australia. (B) The locations of the loggerhead turtle nests monitored during the 2014–2015 nesting season. (C) The locations of the loggerhead turtle nests monitored during the 2015–2016 nesting season. Shaded grey area indicates the section of beach monitored in this study.

recorded. After reading, plots were resurfaced using a rake to obliterate tracks, insuring the same tracks were not recorded on subsequent days. The activity of predators was quantified using the passive activity index (PAI) of *Engeman, Allen & Zerbe (1998)*:

$$PAI = \frac{1}{d} \sum_{j=1}^{d} \frac{1}{Pj} \sum_{i=1}^{Pj} Xij$$

where the *Xij* value represents the number of tracking plot tracks by an observed species at the *i*th plot on the *j*th day; *d* is the number of days of inspection, and *Pj* is the number of plots contributing data on the *j*th day. PAI was calculated for each day throughout the study for statistical comparisons, and at 10-day intervals for graphical presentation of data. Because PAI data failed a Kolmogorov–Smirnov normally test, K–S $d = 0.223, p < 0.01, n = 320$; a non-parametric Kruskal-Wallis ANOVA by Ranks followed by a multiple comparison test was used to test for inter-species and inter-year differences in nest visitation rates.

## Nest monitoring

Once a nest was located it was visited daily throughout the incubation period in order to identify predation events and the tracks of animals visiting nests. Each nest was inspected during the morning (weather permitting) and the number of goanna and fox tracks was recorded. Nest area approximately 1 m² was resurfaced by using a rake after observation. Nest visitation rate was quantified as a percentage by dividing the number of days fresh tracks were found at a nest by the total number of nest inspection days (nest inspection days = total number of times a nest was inspected during the season until hatchlings emerged

from the nest or until it was totally predated). Because goanna nest visitation data failed a Kolmogorov–Smirnov normally test, K–S $d = 0.1442, p < 0.05, n = 121$; a non-parametric Kolmogorov–Smirnov two sample test was used to test for inter-season differences in nest visitation rates.

### Camera traps

Camera traps (Reconyx Hyperfire HC600, Holmen, Wisconsin, USA) were set up to capture images of predators visiting a sample of 12 loggerhead turtle nests (randomly selected) between 6 December 2014 and 27 January 2015 and 30 nests (randomly selected) between 1 December 2015 and 27 February 2016. Camera traps were at each nest for 25 days in 2014–2015 and 30 days in 2015–2016. All camera traps were triggered by motion sensors and could be triggered 24 h per day. Camera traps were positioned 50 cm behind the selected turtle nests, at least 30 cm above ground. Each camera trap had a 1 m² field of view over the nest insuring that any nest visitation by predators was recorded. This enabled information on the frequency, time of day and species to be collected. To compare the relative activity of goannas visiting nests each year and between years, we calculated the per-nest per-day visitation rate for camera trap monitored nests. For each nest the number of independent images (defined as taken at least 20 min apart; multiple images taken within 20 min of each other were classified as a single visitation event) of goannas recorded at that nest was divided by the number of days the camera was set at that nest. Because camera trap per-nest per-day data failed a Kolmogorov–Smirnov normally test, K–S $d = 0.292, p < 0.01, n = 84$; a non-parametric Kruskal–Wallis ANOVA by Ranks followed by a multiple comparison test was used to test for inter-species and inter-year differences in nest visitation rates. Circular statistics were used to analyze the time of day that nests were visited by yellow-spotted goannas and lace monitors.

Non-parametric statistical analysis was performed using Statistica Ver 13.1 (Dell Inc., Rock Round, TX, USA) software, and circular statistical analysis was performed using Oriana Ver 4 (Kovach Computing Services, Isle of Anglesey, Wales, UK).

## RESULTS

### Tracking plots

Monitored tracking plots revealed tracks of two potential egg predators, goannas (lace monitors and yellow-spotted goannas combined as it was not possible to distinguish between the two species on the basis of their tracks alone) and foxes. Only a few dog tracks were identified in tracking plots during the course of the study. However, these dog tracks were most likely made by pet dogs accompanying tourists visiting the beach, and so have been excluded from analysis.

During the first nesting season (5-Dec-2014 until 4-Mar-2015), 21 plots were monitored for 71 days with 466 goanna and 62 fox occurrences recorded (Table 1). During the second nesting season (5-Dec-2015 until 28-Feb-2016), 41 plots were monitored for 89 days with 535 goanna and 70 fox occurrences recorded (Table 1). There were differences in occurrence rates detected between goannas and foxes and years (Table 1), Kruskal–Wallis test: $H(3, n = 320) = 180.065, p < 0.001$. Multiple comparison tests indicated that goanna

**Table 1  Nest visitation events by tracking plots.** Passive activity index (PAI) of goannas and foxes from dune tracking plots during the 2014–2015 and 2015–2016 sea turtle nesting seasons at Wreck Rock beach.

| Nesting season | 2014–2015 | 2015–2016 |
|---|---|---|
| Plots monitored | 21 | 41 |
| Monitored days | 71 | 89 |
| Goannas | | |
| Total events recorded | 466 | 535 |
| Daily PAI | | |
| Mean ± SD | 0.313 ± 0.217 | 0.150 ± 0.104 |
| Median, 25%–75% quartile | 0.286, 0.143–0.423 | 0.125, 0.075–0.200 |
| Range | 0.000–1.048 | 0.000–0.500 |
| Foxes | | |
| Total events recorded | 62 | 70 |
| Daily PAI | | |
| Mean ± SD | 0.042 ± 0.079 | 0.020 ± 0.033 |
| Median, 25%–75% quartile | 0.000, 0.000–0.048 | 0.000, 0.000–0.025 |
| Range | 0.000–0.381 | 0.000–0.175 |

activity was approximately seven times greater than fox activity in both seasons, and goanna activity in 2014–2015 was approximately twice that in 2015–2016, but fox activity was not different between the two seasons. During the 2014–2015 nesting season, goanna activity on the dune front remained relatively constant throughout the season (Fig. 2). Fox activity was generally much lower than goanna activity from December through January, but there was a conspicuous increase in fox activity in February (Fig. 2). In the 2015–2016 nesting season, goanna activity was relatively low in December, increased during January and February and decreased again at the end of February and was lowest in April at a time when most sea turtle nests had hatched. Fox activity remained low and relatively constant throughout the entire season (Fig. 2).

## Nest monitoring

During the 2014–2015 nesting season, 52 loggerhead turtle nests were monitored, and 57.7% of these nests were predated by goannas as indicated by burrows constructed into the nest egg chamber. During the 2015–2016 nesting season, 46 nests were monitored, and 17.4% of these nests were predated by goannas. No fox or other predators were observed to raid turtle nests in either season. During 2014–2015, 520 goanna nest visits (lace monitors and yellow-spotted goannas combined as it was not possible to distinguish between the two species on the basis of their tracks alone) were recorded while in 2015–2016, 343 nest visits were recorded (Table 2). Daily per-nest visits were significantly greater in 2014–2015 than in 2015–2016 (Kolmogorov–Smirnov two sample test, $p < 0.001$) being approximately two times greater (Table 2). Nests that were predated could be dug open for the first time at any time during the incubation period; there was no trend for the first nest attack to be associated with nest construction or nest hatching (Fig. 3).

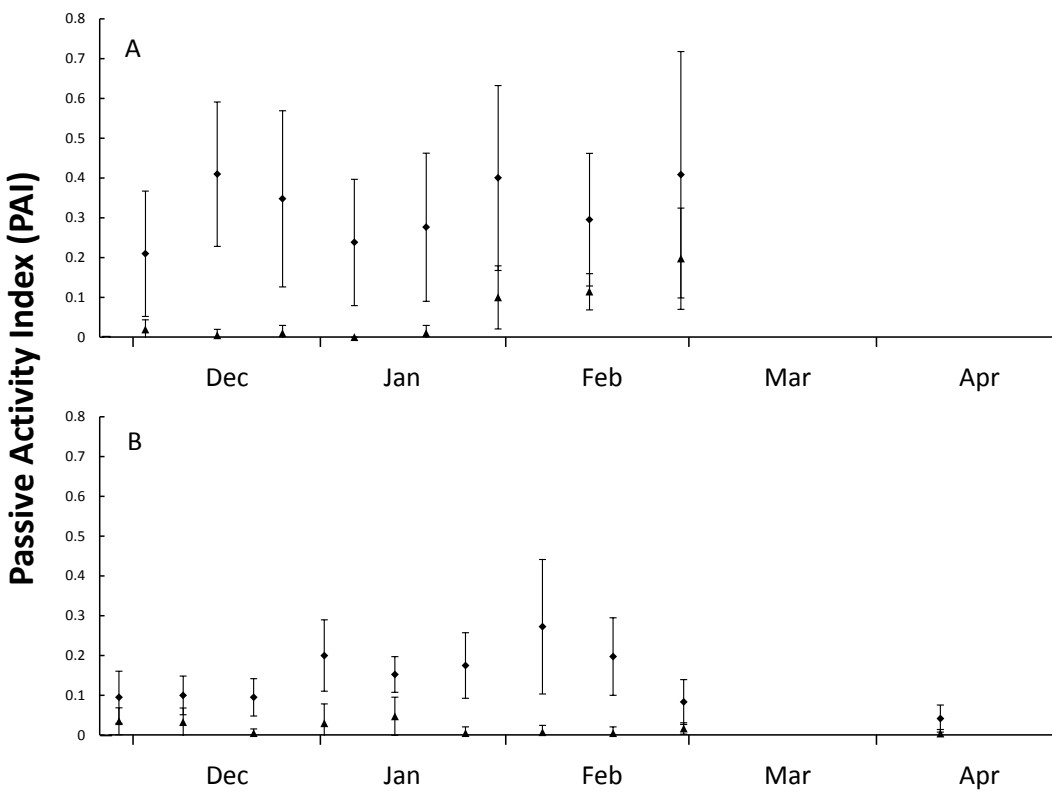

**Figure 2** **Figure of nest predator activity index (PAI).** Passive activity index (PAI, mean ± SD of 10-day intervals) on the front dune at Wreck Rock Beach during the (A) 2014–2015 and (B) 2015–2016 nesting seasons for goannas (diamonds) and foxes (triangles). Data represent the center of 10-day means.

**Table 2** **Visitation rate of goannas based on tracks found on top of nest.** Visitation rate of goannas (based on tracks found on top of nests) at loggerhead turtle nests (visits per-nest per-day) for the 2014–2015 and 2015–2016 sea turtle nesting seasons at Wreck Rock beach.

| Nesting season | 2014–2015 | 2015–2016 |
|---|---|---|
| Nest monitored | 52 | 46 |
| Monitored days | 41 | 80 |
| Total events recorded | 520 | 343 |
| Daily visitation rate | | |
| Mean ± SD | 0.336 ± 0.187 | 0.110 ± 0.107 |
| Median, 25%–75% quartile | 0.312, 0.212–0.476 | 0.087, 0.022–0.175 |
| Range | 0.000–0.882 | 0.000–0.455 |

## Camera traps

Images from camera traps showed that goannas were the only predators to visit monitored nests; no images of foxes or wild dogs were recorded. All of the monitored nests had at least one image of a goanna visit during the deployment period, with 55 nest visitation events being recorded in the 2014–2015 nesting season, 47 (85.5%) of these visitation events were made by yellow-spotted goannas and only eight (14.5%) were made by lace monitors (Table 3). The overall per-nest per-day visitation rate was 0.157 for yellow-spotted goannas

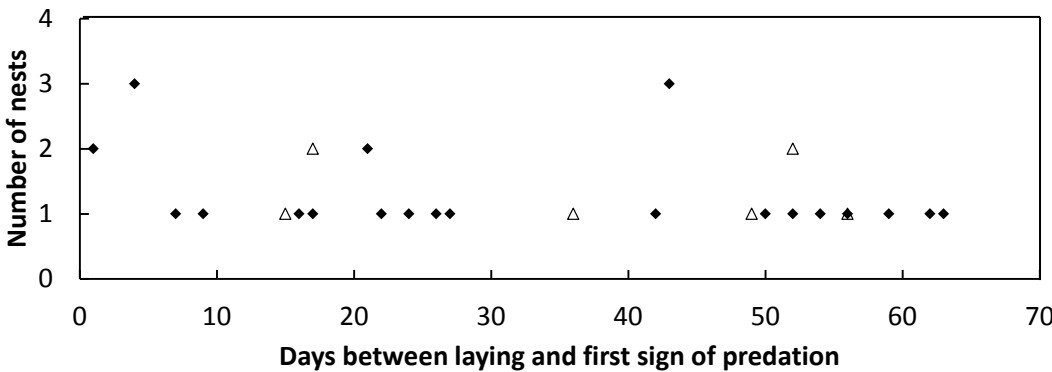

**Figure 3** **Plot of the frequency of nest predation events against the time since nest construction and first goanna predation event for loggerhead nests laid.** Frequency of goanna first predation events in relation to the time since nest construction for loggerhead nests laid during the 2014–2015 (solid diamonds) and 2015–2016 (open triangles) nesting seasons at Wreck Rock beach.

and 0.027 for lace monitors (Table 3). Despite all camera traps being deployed by 20 December 2014, only two goannas appeared at nests in December 2014, but activity at nests increased sharply from the beginning of January 2015 (Fig. 4A). Eggs were seen to be consumed on 17 occasions (14 yellow-spotted goannas, 3 lace monitors). Yellow-spotted goannas were seen to open a nest for the first time on 17 occasions, but lace monitors were only ever seen to visit nests that had already been opened. In the 2015–2016 nesting season, 107 goanna nest visiting events were captured, 87 (81.3%) of these visitation events were made by yellow-spotted goannas and only 20 (18.7%) were made by lace monitors (Table 3). The overall per-nest per-day visitation rate was 0.097 for yellow-spotted goannas and 0.022 for lace monitors (Table 3). Eggs were seen to be predated by yellow-spotted goanna on 6 occasions. No lace monitors were seen consuming eggs in this season. There were difference in visitations rates between monitored groups, Kruskal–Wallis test: $H(3, n = 84) = 26.826, p < 0.001$; with multiple comparison tests indicating that yellow-spotted goanna visitation rate was greater than lace monitors in both seasons, but there was no difference in inter-season visitation rate in either species.

Goannas visited nests at any time of the day between 8:00 and 18:00 (Figs. 4A and 4B). Combining data from both seasons, and plotting the data separately for yellow-spotted goannas and lace monitors revealed that yellow-spotted goannas had a bi-modal nest visitation pattern, with a peak in activity in the morning between 7:00 and 11:00, 8:57 ± 1:10 (mean ± circular standard deviation) and again in the afternoon between 13:00 and 16:00, 15:10 ± 0:50, while the most frequent time for visits from lace monitors was in the afternoon the late afternoon between 15:00 and 17:00, 15:46 ± 0.50 (Fig. 5). A Watson–Williams $F$-test ($F_{1,154} = 11.792, p < 0.001$) confirmed that when considering all nest visitation data, the mean time of lace monitor visits (13:38 ± 2:53, $n = 29$) was later than yellow-spotted goanna visits (11:27 ± 3:00, $n = 129$).

An entire nest opening sequence was recorded on 23 Jan 2015. A large yellow-spotted goanna first began digging at 14:12 (Fig. 6A). It reached the egg chamber and consumed the first egg at 14:28 after 16 min of continuous digging activity (Fig. 6B). Turtle eggs were

**Table 3   Nest visitation rate by camera traps.** The nest visitation rate (per-nest per-day) recorded by camera traps at nests during 2014–2015 and 2015–2016 sea turtle nesting seasons at Wreck Rock beach.

| Nesting season | 2014–2015 | 2015–2016 |
|---|---|---|
| Nests monitored | 12 | 30 |
| Monitored days | 25 | 30 |
| Yellow-spotted goannas | | |
| Total events recorded | 47 | 87 |
| Visitation rate | | |
| Mean ± SD | 0.157 ± 0.247 | 0.097 ± 0.116 |
| Median, 25%–75% quartile | 0.060, 0.040–0.160 | 0.050, 0.033–0.133 |
| Range | 0.040–0.920 | 0.000–0.467 |
| Lace monitors | | |
| Total events recorded | 8 | 20 |
| Visitation events per nest per day | | |
| Mean ± SD | 0.027 ± 0.049 | 0.022 ± 0.029 |
| Median, 25%–75% quartile | 0.000, 0.000–0.040 | 0.000, 0.000–0.033 |
| Range | 0.000–0.160 | 0.000–0.100 |

swallowed intact, one at a time, by the goanna rather than being opened and having their contents licked out (Fig. 6C). This goanna stopped feeding and left the nest at 16:56 after almost 2.5 h of feeding and having consumed eight eggs.

# DISCUSSION

Nest predation decreases the recruitment of hatchlings and has become an important challenge for the conservation of egg-laying reptiles (*Leighton, Horrocks & Kramer, 2010*). Hence, understanding the activity of predators adjacent to endangered reptilian species breeding aggregations is important for designing conservation strategies. The daily checking for predator tracks on nests and the deployment of tracking plots and camera traps allowed us to continuously monitor activities of nest predators adjacent to a loggerhead turtle nesting beach. There were two significant results from the study that provide new insights into goanna predation of sea turtle nests. First, camera trap data indicated that yellow-spotted goannas are the most frequent visitors and predators of sea turtle nests at Wreck Rock beach and were the only species observed to open nests, suggesting they are the main cause of nest predation. Second, the nest predation rate and activity of goannas on the nesting dune varied by a factor of two between the two seasons that we studied.

## Predator activities at nests

Camera traps allowed us to explore the loggerhead turtle nest predator species, predation time and behavior of predators while at nests. Yellow-spotted goannas were the most frequent visitors and predators of sea turtle nests in this study. Large adult yellow-spotted goannas have the ability to dig up sea turtle nests and swallow turtle eggs intact, suggesting future management strategies should be targeted at these individuals. Indeed, no lace monitors were observed to open sea turtle nests directly, they were only observed predating nests that had already been opened by yellow–spotted goannas. Hence, lace monitors appear
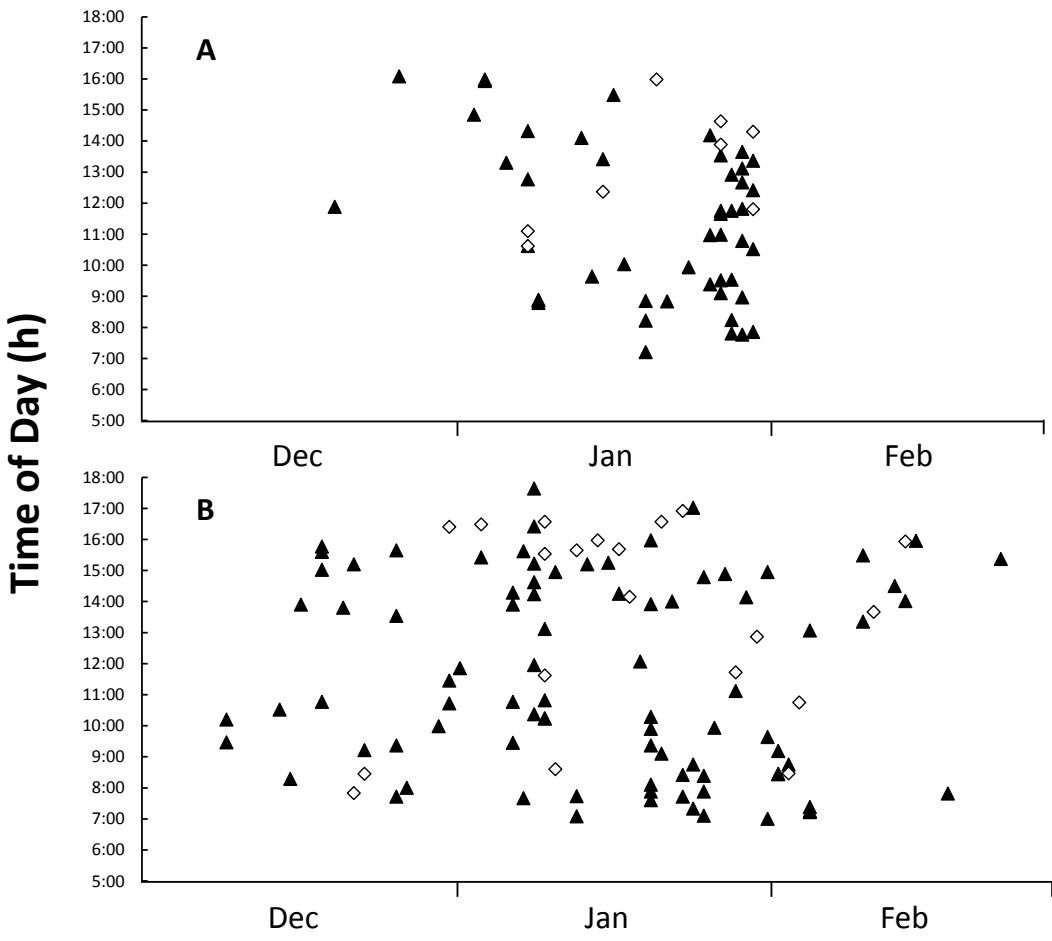

**Figure 4** **Figure of time and date of goanna appearances at loggerhead turtle nests as determined from camera trap records.** Time and date of goanna appearances at loggerhead turtle nests as determined from camera trap records. Triangle, yellow-spotted goannas, Diamond, lace monitors. (A) Three hundred camera days (12 cameras set for 25 days each) during the 2014–2015 season. (B) Nine hundred camera days (30 cameras set for 30 days each) during the 2015–2016 season.

to be opportunistic nest predators on this beach. Lace monitors are frequently arboreal and are equipped with long, recurved claws that facilitate climbing (*Cogger, 1993*). Such claws are not particularly useful for digging and may explain why they did not open nests. Using GPS tracking methodology, *Lei & Booth (2015)* reported yellow-spotted goannas use the beach dunes more than lace monitors and are therefore more likely to predate sea turtle nests than lace monitors. Moreover, camera traps did not record foxes at nests, and no fox tracks were observed over nests during this study indicating that the fox baiting program deployed by park managers is currently effective at inhibiting fox predation of sea turtle nests at Wreck Rock beach.

Although camera trap records indicated that sea turtle nests were visited by yellow-spotted goannas at any time of day between 7:00 and 17:30, visits were most frequent in the morning and afternoon with a distinct lull during the middle of the day. This reflects the general activity pattern of yellow-spotted goannas as recorded by GPS tracking
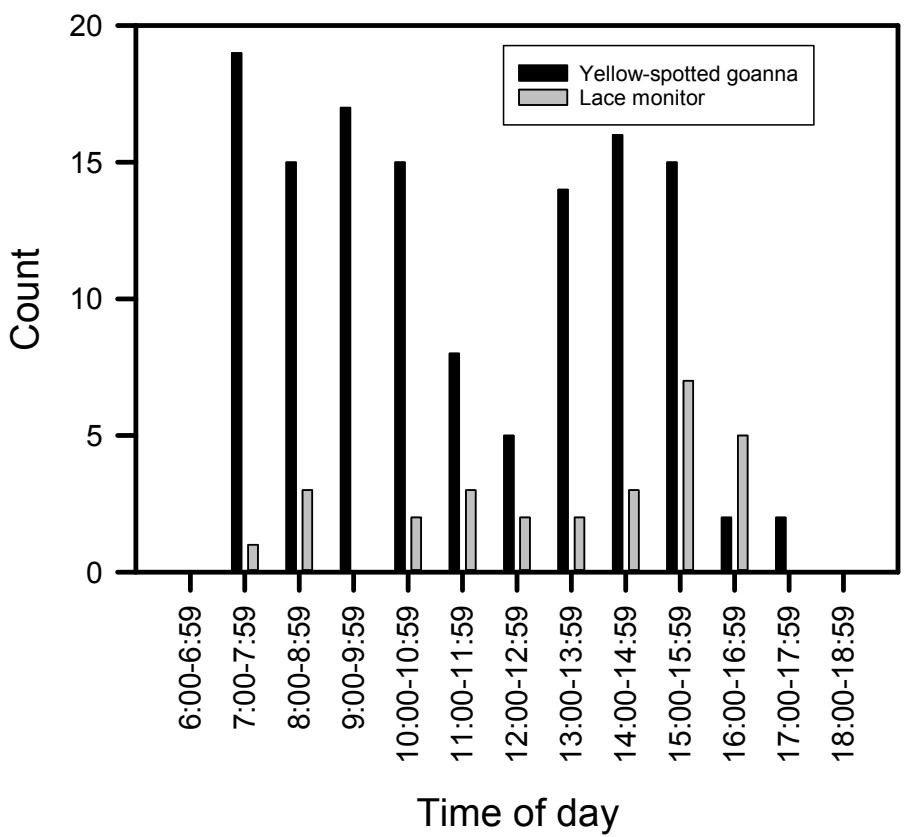

**Figure 5** **Plot of the number of images of goannas against time of day.** The number of images of goannas taken by camera traps set at loggerhead turtle nests at Wreck Rock beach in relation to time of day that images were recorded.

data (*Lei, Booth & Dwyer, in press*). It would appear that the midday heat suppresses the foraging activity of yellow-spotted goannas, and this may be particularly so in the beach dune area there are no trees to provide shade. In contrast, although the data is far less numerous, lace monitors had a single peak in sea turtle nest visiting activity, and this was late in the afternoon, typically after the peak afternoon yellow-spotted goanna nest visiting time. Hence, lace monitors may arrange their nest visiting times to avoid interacting with yellow-spotted goannas. Further investigation of this possibility is needed.

*Doody et al. (2014)* and *Doody et al. (2015)* reported that yellow-spotted goannas can dig warren complexes that required removal of sand from up to 3 m deep and that both males and females contribute to warren excavation. Hence, the job of digging into a sea turtle nest which is comparatively shallow (40–80 cm), should be relatively easy as evidenced by it requiring only 16 min of digging to gain access to eggs in one of our monitored nests. Our camera trap images indicated that yellow-spotted goannas normally dug into the nest at an angle from one side of the nest to reach the nest chamber rather than digging a hole vertically downwards from directly above the nest. Hence, when covering a nest

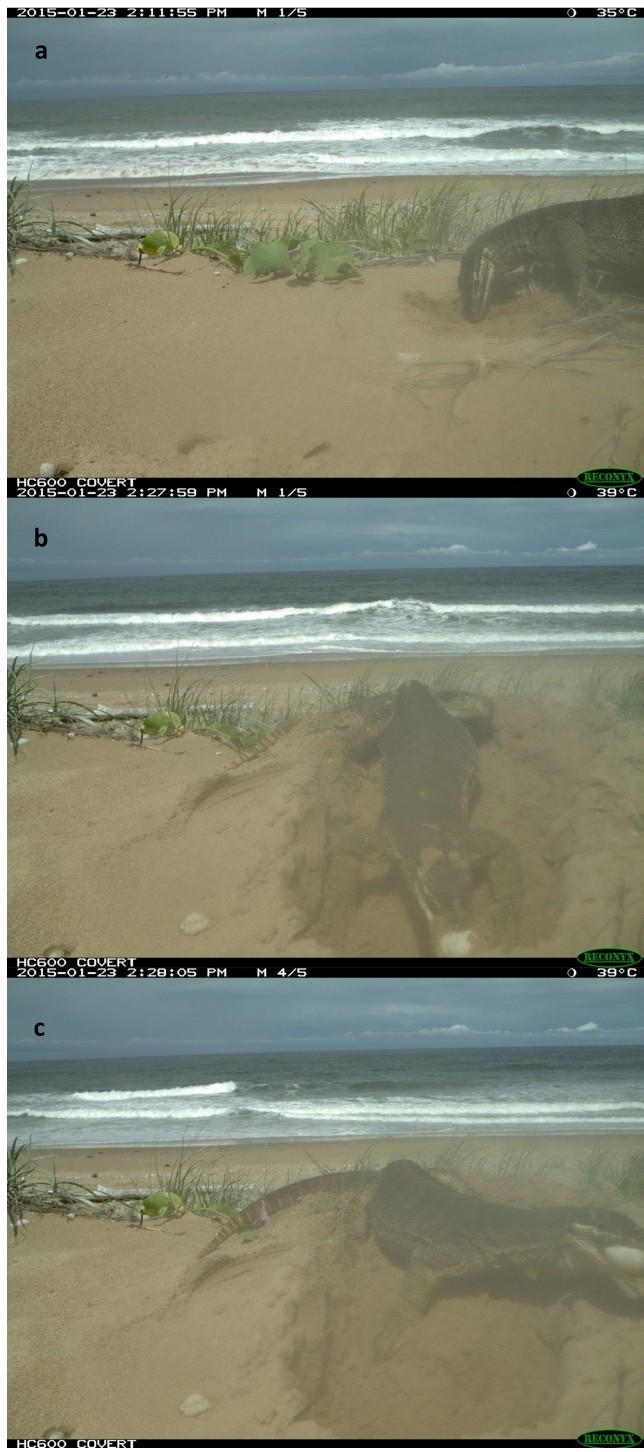

**Figure 6  The photos of a yellow-spotted goanna opening and consuming eggs from a loggerhead turtle nest.** A Yellow-spotted goanna opening and consuming eggs from a loggerhead turtle nest on 23-Jan-2015. Photos were captured by a camera trap. (A) Start of digging, (B & C), removal and consumption of the first egg. For full sequence, see video in the Supplementary information.

with mesh as a management strategy used to deter nest predation, the mesh must be relatively large in area (at least 1 × 1 m) to prevent yellow-spotted goanna burrowing into the nest (*Lei & Booth, in press*). Turtle nest predation rate is likely dependent on cues left by the female turtle (e.g., visual, tactile, and olfactory), and many predators have the ability to detect these cues (*Van der Wall, 1998*; *Van der Wall, 2000*; *Geluso, 2005*; *Leighton, Horrocks & Kramer, 2009*). Goannas use their forked tongue to transfer olfactory cues to the specialized chemosensory Jacobson's organ and so are adept at using olfactory cues to find prey (*Blamires & Guinea, 1998*; *King & Green, 1999*; *Vincent & Wilson, 1999*). We found that once a turtle nest was opened, this nest was continually predated over subsequent days by multiple yellow-spotted goannas.

We suspected that goannas might attack sea turtle nests more frequently immediately after their construction, or after hatching at the end of incubation. These expectations were based on the idea that sand disturbance and the smell of the female and or newly laid eggs around the sand might give clear clues to foraging goannas immediately after nest construction, and that the smell of egg fluids released during the hatching process might also attract goannas at the end of incubation. This was not what we observed; a nest was equally likely to be attacked for the first time at any time during incubation. We do not know why this is the case, particularly as goannas crawled over the top of some nests several times during incubation without attacking them, and then at a later date these nests were attacked. One possibility might be that ghost crabs (*Ocypode ceratophthalmus* and *O. cordimanus*) which are numerous on the nesting beach and frequently burrow into sea turtle nests, cause the release of 'incubating egg odor' which then attracts goannas.

## Predator activity

Based on the PAI analysis of tracking plot data, the activity of goannas was higher than that of foxes, suggesting that goannas are the main predator of sea turtle nests at Wreck Rock beach, a conclusion also supported by nest track and camera trap data. We found that all of our monitored nests were visited by goannas and that between 17% (2015–2016) and 58% (2014–2015) of nests were opened by yellow-spotted goannas. Goanna predation of nests had previously been reported as greater than 50% at this beach (*McLachlan et al., 2015*). It is unclear whether goanna predation of sea turtle nests was this high at Wreck Rock beach during pre-European settlement times or whether more recent perturbations have led to increased nest predation in recent times. During the 1970s–1990s goanna predation of sea turtle nests at this location was not detected, but fox predation of nests was high, 90% of nests being predated in the 1970's and up until 1987 (*Limpus, 2008*). From 1987 onwards, a fox baiting program reduced fox predation on sea turtle nests to negligible levels (*Limpus, 2008*). Goanna predation of sea turtle nests was first reported in the 2003–2004 nesting season when two nests were predated (*Limpus, 2008*), and since then goanna predation of sea turtle nests has increased so that over 50% of sea turtle nests were being attacked by goannas in the 2013–2014 season (*McLachlan et al., 2015*). Hence, the reduction in fox numbers may have also resulted in an increased recruitment of yellow-spotted goannas (because red foxes probably also predated yellow-spotted goanna nests) to historically high levels.

However, before European settlement and the introduction of foxes, hunting of goannas by native people may have kept the density of goannas on the frontal dunes at a low level.

Goanna activity in 2014–2015 was twice as high as in 2015–2016, as was the nest predation rate. This suggests that nest predation is positively correlated with goanna activity. *Maulany (2012)* reported that olive ridley turtle nests suffered 100% predation by monitor lizards at a beach adjacent to Alas Purwo National Park, Banyuwangi (East Java), Indonesia, which had high monitor lizard activity (PAI = 1.27 in 2009, 1.41 in 2010). This finding also suggests that goanna activity on dunes is a good predictor of intensity of goanna predation on sea turtle nests.

Fox activity increased at the end of the 2014–2015 nesting season. Typically the park mangers fox bait twice during the sea turtle nesting season, once in early December and again in early February. In 2014–2015 the February baiting was missed, so any foxes that might have moved into the beach area after the December baiting were not removed. However, in the 2015–2016 season, the early February fox baiting probably maintained fox activity at low levels.

The goanna predation rate of sea turtle nests in 2014–2015 was twice that in 2015–2016, and it correlated with an increase in goanna activity on the dune. The nest visitation rate by recording tracks on nests in 2014–2015 was nearly twice that in 2015–2016. These results suggested goanna activity on the dune in 2014–2015 was higher than in 2015–2016. However, it remains unclear why goanna activity and sea turtle nest predation rate varied so greatly between the two nesting seasons. Because of the strong inter-annual differences in predator indices over two years, additional years of research are needed to determine the long-term average predation rate and its implications for turtle hatching success.

## Implications for management

*Lei & Booth (in press)* compared different methods of directly protecting sea turtle nests against goanna predation and found that deploying the plastic mesh on the top of turtle nests was the most effective and economic way. Combined with our observations of digging behaviour of yellow-spotted goanna captured on camera traps, we suggested that plastic mesh needs to be at least $1 \times 1$ m to prevent yellow-spotted goannas digging into the nest chamber. In addition, camera trap data indicated turtle nest predation activities happen any time between 7:00 and 17:00, suggesting turtle nest management should be deployed in the early morning following the night that nests are constructed. More management strategies such as temporary removal of large male yellow-spotted goannas or egg relocation should be investigated in the future to counteract the loss of sea turtle nests to yellow-spotted goanna predation.

## ACKNOWLEDGEMENTS

This work would not have been possible without the help of Nev and Bev McLachlan's Turtle Care Volunteers organization, the Burnett Mary Regional Group and WWF Australia.

### Funding

This work was supported by a grant from the Nest to Oceans Turtle Protection Program, a program jointly funded by the Australian and Queensland Governments. The funders had no role in study design, data collection and analysis, decision to publish, or preparation of the manuscript.

### Grant Disclosures

The following grant information was disclosed by the authors:
Nest to Oceans Turtle Protection Program.
Australian and Queensland Governments.

### Competing Interests

The authors declare there are no competing interests.

### Author Contributions

- Juan Lei conceived and designed the experiments, performed the experiments, analyzed the data, contributed reagents/materials/analysis tools, wrote the paper, prepared figures and/or tables, reviewed drafts of the paper.
- David T. Booth conceived and designed the experiments, performed the experiments, analyzed the data, contributed reagents/materials/analysis tools, prepared figures and/or tables, reviewed drafts of the paper.

### Animal Ethics

The following information was supplied relating to ethical approvals (i.e., approving body and any reference numbers):

All work was approved by a University of Queensland Animal Ethics Committee (permit #SBS/352/EHP/URG) and conducted under Queensland Government National parks scientific permit #WITK15315614.

### Data Availability

Lei, Juan (2017): Combined tracking plot, camera trap and nest monitoring data. figshare.

https://doi.org/10.6084/m9.figshare.5032478.v1.

### Supplemental Information

Supplemental information for this article can be found online at http://dx.doi.org/10.7717/peerj.3515#supplemental-information.

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
