# Peer review of "Who are the important predators of sea turtle nests at Wreck Rock beach?"

_PeerJ, doi:10.7717/peerj.3515_

## Round 0.1 · original submission · Minor Revisions

Both reviewers felt that your study makes a useful contribution to the literature on predation on sea turtle nests, but both also felt that the presentation needed to be strengthened. My own reading concurs with those views. Although many changes are suggested, most are relatively minor and would not change the findings. Therefore, my decision is to request minor revision.

I have provided a copy of your pdf with problems highlighted and suggestions for changes inserted in comments. To facilitate your revision, I used the pdf which had already been annotated by Reviewer 2. You will be able to distinguish our comments because mine are a distinguished by orange color and my initials (DLK).

Below, I highlight several issues. In your response to the revision, please include all these comments as well as those of the referees. In addition, you should indicate your response any substantial comments by the reviewer and myself on the manuscript itself if they are not mentioned below. You do not need to respond explicitly to changes in spelling, grammar, punctuation, word order, and format unless you disagree and decide to retain the original text.

Editor's Comments

As I understand it, you have three measures: track occurrence at nests, camera records at nests, and track occurrence in the general area. On my first reading, I did not recognize the use of tracks at nests and was confused by some apparent contradictions in the Methods and Results. I believe that the problem is the nest measure was not mentioned in the abstract and received extremely brief mention in the Methods (L109ff). No detailed methodology was described (and it should be), and the measure was given the same name as the camera data (and the terms for each measure should be different).

I would like you to reconsider the term 'passive soil plots'. I do not find it very revealing and do not find a clear precedent for this term. If I am correct, it would not help searchers using key words to find your study. 'Soil plot' sounds as if it might refer to plant ecology, and 'passive' seems redundant because an 'active soil plot' doesn't make sense. Furthermore, you add ambiguity by using the alternative 'passive sand plot' in some places. Engeman, whom you cite, refers to these as 'tracking plots'. A search on Web of Science seems to indicate only his research uses this term. Leighton's papers (of which I should admit I was a co-author as his doctoral supervisor), which you also cite, use the term 'track pad'. This term is used by a few more authors in the Web of Science (but also applies to computer components and military tanks). Whatever you call it, when you first introduce the concept, around L80, you need to describe the plot as well as the PAI.

Reviewer 1 suggests that you examine the effect of nest age on predation. You could also consider seasonal effects. From my work with Leighton, I know that such an analysis is complicated, and I do not insist on it, but you should carefully consider the suggestion.

I looked at the supplementary information file and did not find it very clear, although my inspection was rather brief. Please check that an independent reader would be able to understand your headings and recreate your data set from this file. In addition, there is a column for treatment, which appears to be predation reduction treatment. If you were carrying out other treatments on the nests reported here, it is crucial that you indicate this and address potential implications for your findings. If some of the same data are reported in other publications, you have an obligation to indicate the relationship between this manuscript and the others.

L57 (and elsewhere, if applicable). Provide scientific names for all species at first mention.

L66ff. I strongly agree with the reviewer that you have not clearly identified the knowledge gap that your study addresses. By the criteria of PeerJ, this does not have to be a substantial gap, and replication of previous studies is acceptable. Even just seeing whether the pattern has changed in recent years could be an adequate goal. However, it is critical to give credit for all previous research and clearly indicate any findings relevant to your study. Your Introduction implies that your findings were already established, and the Discussion implies that this was done by Limpus 2008 as well as the cited study by McLachlan et al.

L91ff. The manuscript would be easier to follow if your objectives were given in the same order as the Methods and Results. Furthermore, the order of findings in the Abstract should follow the same logical order as the Methods and Results. Your goal of examining the relationship between goanna activity and nest predation implied to me something more than the non-statistical comparison between two years that you provided (for example, comparisons between areas or different parts of the season). Consider whether this is really an appropriate objective for your study and whether you have adequately met this goal.

L117ff. Some relevant information on the camera traps is missing. Did they record at fixed intervals or were they motion sensitive? Did they record throughout the night as well as the day? Is there anything else the reader should know about them - how far away, size of the field, . . .?

L123 I found your description of the visitation rate calculation somewhat ambiguous. The statement about all nests monitored in the season seems to imply that the data were aggregated before the visitation rate was calculated. Revise to make clearer. Providing units would help. Starting with the definition of nest visitation rate and then defining camera trap days might help the reader follow more easily.

There are some errors and ambiguities as well as inconsistent formatting in the references. Please check all references carefully.

Table 1. I agree with Reviewer 2 that Table 1 should provide a summary rather than values for individual nests. Individual records should be in the supplementary data. I realize that these records are somewhat aggregated in that you have provided seasonal totals rather than weekly values. In addition, the PAI values for activity on the dunes are provided in Fig. 4, so you should not repeat that information. However, it seems like summaries of visitation rates from both your track records and camera records would be appropriate. You need to consider whether aggregating by time or nest is most appropriate. It should be consistent with your general reporting (i.e. are you focused on the variation among years, weeks or nest or all of these?). In addition, looking at the data suggests that the among nest patterns are not normally distributed. If this is the case, medians with quartiles and ranges would be more appropriate than means and standard deviations.

Fig. 1. I do not agree with the reviewer that you should make the symbols of nest locations smaller. That would still create difficulties for readers. Perhaps you could use separate panels for the two years. The map of northeast Australia could be greatly reduced and provided as an inset. Also, the scale does not appear to correspond to your description. You say you observed nests and goannas 2 km north and south of the rock, but the inset scale seems to indicate a distance greater than 3 km.

Fig. 4 refers to an online video of nest predation, but it is not clear where this is. It does not appear to be listed in the PeerJ information.

I have provided detailed comments on the pdf in an attempt to increase conciseness and remove errors in spelling, grammar and punctuation. There are many sentences where a switch to active voice would further help the readability. It might be a good idea to review the use of commas before undertaking your revision, in case I missed some mistakes and to avoid errors in your revised text. Also, a spell check would have caught some of the errors.

Reviewer 1 ·

Basic reporting

I believe that the writing should be improved, mainly in the discussion section. I suggest that you rewrite your discussion in order to make it clearer and less speculative.

In order to make the discussion clearer, I suggested that you cut repetitive paragraphs such as found in the lines 291 to 302; and remove excessive repletion of results; some examples include lines 304-308.

The structure of the manuscript is on an acceptable format. The figures and tables are relevant to the content of the manuscript. Although, figure 4 may be improved by adding labels that explain the meaning of solid and dashed lines direct to the figure. All the relevant raw data was made available.

Experimental design

This manuscript is within the Aims and scope of the journal. The methods descriptions allowed the replication of this study. The research question is well defined, however, to achieve a high quality paper I believe that the results (and consequently the discussion) should be improved. I think that manuscript will benefit by answering the following questions:

How nest predation impacts on nest success (e.g. average number of viable hatchlings found in predated nests vs undisturbed nests)?

Is there any difference on predation rate related to the age of the nests? Hypothesis: Recently build nests provide stronger olfactory cues to goannas, when compared to older nests.

A statistical analysis of the data from camera trap records may reveal a temporal difference in nests visits by yellow-spotted goanna and lace monitors.

Validity of the findings

As mentioned above, I believe that the quality of the findings of the manuscript may be easily improved by answering some additional questions.

Please find below some other suggestion regarding the findings and discussion:

Some parts of the discussion are very speculative, examples: 285-289

Authors should consider other options to explain the differences predation rate by yellow-spotted goanna and the lace monitor. The low nest predation by lace monitors may be not related to their lack of ability in excavate a turtle nest (considering that lace monitors are able to excavate termite mounds in order to lay their eggs), but they may be less able to find a turtle nest.

I suggest that the authors improve the comparison of their findings to the Maulany (2012) findings.

Additional comments

The manuscript deals with the predation of loggerhead sea turtle nests by goanna along the Wreck Rock beach in Southeast Queensland, Australia. Although I do agree with the authors that their findings help to improve the understanding of nest predation by native species of goanna, which may lead to better management actions, I think that this manuscript should be significantly improved prior its acceptance in PeerJ. There are several underlying issues that should be addressed by the authors mainly in the results and discussion sections.

Reviewer 2 ·

Basic reporting

The paper is generally well written, appropriately referenced and logically structured. I've made suggestions to help the authors improve clarity in the attached annotated manuscript file. In particular, I think Table 1 should be replaced with a table that summarizes their three different predator activity data sets separated by species and year. Also, the information on the IUCN status of loggerhead turtles is out of date and needs to be made specific to the sub-population. I don't see any statements related to data sharing in the manuscript.

Experimental design

This is a primary research manuscript that is well suited for PeerJ. The research question is well defined (although the wording of the study aims could be improved in the introduction). It is highly relevant to sea turtle conservation in Australia and is meaningful from a basic research and management viewpoint. I've suggested improvements in terms of identifying the specific knowledge gap filled by the study. The authors do a really nice job of collecting three complementary data sets on nest predator activity, which allows them to reach a strong conclusion regarding the current relative importance of two species of native goanna and introduced foxes in this loggerhead turtle rookery. The methods are suitable and well described but I have asked for additional details in a couple of places.

Validity of the findings

The data appear robust and the results of the study valid. I recommend that the authors acknowledge to a greater extent that only two years of data limits the strength of inferences/management recommendations that can be made. I've also asked for additional detail on how tracks were recorded to calculate passive activity indices, which I believe is important.

Additional comments

It was a pleasure to review this study. Overall, the authors have done a nice job of collecting complementary data sets to address an interesting and important question for this loggerhead rookery. I have made a number of recommendations that I hope will help improve the manuscript and make it suitable for publication (see annotated pdf).

Annotated reviews are not available for download in order to protect the identity of reviewers who chose to remain anonymous.

---

## Round 0.2 · Minor Revisions

The manuscript has substantially improved. As recommended by a reviewer, there is more acknowledgement of the inter-annual difference. However, this acknowledgement is not supported by statistical analysis, which I believe is needed. In addition, there are quite a few errors in grammar and redundancies that could be eliminated. I am therefore returning it for minor revision.

Editor's Comments

Unlike the previous version, key words are not mentioned. I checked key words on the previous version. You can delete key words that are included in title and abstract. However, I suggest adding 'track pad' as well as the families of the turtles, goannas and foxes.

L44. I find it awkward to assert that the study suggests strong among-year variation in abundance indices. This is your result, not an inference (unless you are trying to extend it to more years). I suggest that you add information earlier in the abstract to indicate the between year variation in nest visitation from tracks and cameras, and modify your conclusion to the suggestion that goanna abundance varied strongly between years.

L54. I happened to notice that you cited an article on bluebird incubation as evidence for factors influencing turtle hatching success. I didn't read the article, but the abstract made no mention of turtles. Are you sure this is an appropriate citation?

L76. The last part of this sentence is redundant.

L99. Although not an actual error, it seems odd to use a 1-page article on radioisotopes from a conference report in the 1970s as a citation for the difficulty in estimating populations in general. As it is a well-recognized issue, can't you find a more general and recent reference?

L100. The introduction of the concept of PAI and tracking plots doesn't quite work because you do not explicitly link the two concepts. The sentence on PAI needs to specify that it uses data from tracking plots with a brief indication of what they are. The sentence will need to be divided so that it is not too long. I have suggested a change on the pdf.

L189ff. The presentation of your dune activity information is not completely clear. I presume that the n's are the number of tracking plot-days with recorded activity. This needs to be explicit. It would be useful to indicate the total number of plot-days to put this in perspective. When you provide PAIs, you also need to indicate where the SE came from. I suppose this might be based on weekly values. In such a case, you could state something like 'weekly PAI was 0.31 ± 0.03 (mean ± SE)'. However, I think that SD rather than SE is usually recommended for descriptive data and the n should be provided (number of weeks?). Since you make a point concerning the between-year differences, I think it would be appropriate to have statistical support for the difference. This could also apply to the difference between goannas and foxes.

L206, 212, 229. Are there statistically significant differences between years in predation, visitation, and camera trap rates? Would there be value in providing SD and range (or median, quartiles, and range, if not normally distributed) for per nest visitation and camera trap observations?

References. Despite a request to carefully check your references, numerous spelling and other errors remain. For the reports of symposia, you need to include editor(s), publisher, and place.

Table 1. The table needs to include a statement of what the variation represents in rows 4 and 6. For example, following the variable definition in each line write mean ± SD. You should space around the ± symbol. Also, please check the appropriateness of your measure of central tendency and variation (mean vs. median, SD vs. SE vs. quartiles and range).

Fig. 1C. Shading for rock is missing.

Fig. 2. Caption should define error bars. It would probably be a good idea to indicate a break in the solid lines using // because sampling occurred at a longer interval.

Supplementary data. For Tracking Plot data, both files refer to 2014-2015 in the top line. Even though the file names are different, this error could result in confusion. Similarly, for the Nest Monitoring files, both refer to 2015-2016 in the top line. Note also a shift in date formatting within the Nest Monitoring files. To reduce the number of separate files, I imagine it would be reasonable to include a single file using different tabs (clearly labeled) for the different data sets.

I have provided an annotated pdf to indicate the grammatical and stylistic changes needed.

---

## Round 0.3 · Minor Revisions

Unfortunately, there are still some problems with your manuscript. Some of your previous responses have not explicitly addressed all the suggestions provided. It will save you and me considerable time if you carefully check that you have addressed each point raised and included how you addressed that point in your response.

1) I asked if there were significant differences in predation, visitation, and camera trap rates and if you thought it was worth providing mean, SD and range or non-parametric equivalents, if appropriate. You replied that you had added SD and range and had provided statistical comparisons. Unfortunately, these additions were incomplete and sometimes unclear.
• You did not include range in the descriptive statistics.
• You did not indicate in your response whether or not you had checked that parametric descriptive statistics and statistical tests were appropriate.
• Although you provided p-values, you did not provide the test statistic, misspelled Student's t-test, sometimes failed to include the name of the test, and left it to the reader to try to work out the degrees of freedom from previous information on sample size. It is standard practice in reporting statistics to include the name of the test, the test statistic and d.f. in addition to the p-value.
• The name of the test does not have to be provided in Results if it is clearly provided in the Methods. It would be appropriate to include a statement in the methods for each of your measures specifying the statistical test used, indicating that you checked that the data were appropriate for a parametric test (or using a non-parametric test otherwise).
• The presentation of the statistics in parentheses on L192-194 is quite awkward. I don't see a reason to use parenthesis here or to start the sentence with a number. Please revise. It might help to look at some other research papers as models to see how descriptive data and multiple statistical comparisons can be more concisely and clearly presented.
• In the nest monitoring section, you reported predation rates but not whether the years were significantly different.
• For time of day of visits to nests, I believe that circular statistics are usually required. This may not be a major problem because the visits were strictly diurnal. In addition, the use of parametric statistics would be almost certainly incorrect when the distribution is bimodal. In fact, comparing the mean time of day in such a case may be quite misleading. Please consult a statistician regarding the best way to compare the activity times of your species.

2) Your methods state that tracking plots were measured from December to March, but the figure shows that December is included in your data, but most of March is not. Because of the vagueness of the expression 'from date x to date y', it would be more precise to provide the actual date range in each year in the Methods somewhere around L136.

3) Table 1. You have incorporated my directions into the heading! Please remove that and see suggestions on pdf to clarify the table.

4) Figure 2 has several problems.
• The error bars in the figure should be changed to SD to match the descriptive statistics for your other data. (The data should be medians and quartiles if the data are not normally distributed.)
• The caption is awkwardly written and not very clear. I have suggested an alternative on the pdf, but it may need to change depending on other points raised below.
• You used the term 'track activity' instead of 'passive activity'. I presume that this was unintentional.
• The use of 10-day means need to be mentioned in the caption. Are the points centered on the middle of each 10-day interval?
• If I understand correctly, all points through late February are based on the 5 days preceding and following the point. However, no data were collected between early March and early April. Thus, drawing a line between these points is not appropriate nor is describing it as a longer interval which implies that the other points were separated by intervals during which no data were collected. If my understanding is correct, the line should be removed. If you want to include the shading to indicate the gap, it should indicate the actual gap in sampling, which I presume would last from 5 days after the February point to 5 days before the April point. However, removal of the line would be sufficient.

5) Key words are used to facilitate searches by terms that are not included in the title and abstract, such as alternative expressions for major concepts that might be used by other researchers. This is the reason that I suggested you include 'track pads' in your key words. You replied that you had added key words but you did not state that you had not added track pad, using the term 'tracking plot' that was already in your abstract. If you had a good reason to not include track pad, I could have accepted your argument or tried again to explain my reasoning. However, ignoring the comment forces me to write again to see if there was a good reason, a misunderstanding, or you had simply missed the comment.

6) There are still a few mistakes in the references. Those I detected are highlighted, but you should recheck them all. Put 'p.' before page numbers in proceedings and edited volumes.

7) I have suggested a few changes in other figure captions and main text.

---

## Round 0.4 · accepted · Accept

Following the most recent modifications, I now consider the manuscript ready for publication. I am glad that the statistical analysis now takes into account the distribution of the data, but sorry that I had to be so insistent to make it happen. It is also unfortunate that Smirnov has been consistently misspelled, considering the number of times I had to point out similar careless errors on previous versions. While all of us make mistakes in our writing and can miss typos, the number of errors in various versions of this manuscript was excessive. Taking extra care with revisions would reduce work for you and journal staff, as well as the journal reviewers and editors who are volunteering their time.

L37 Abstract should be 'approximately seven times' to agree with Results
L150, 163 (twice), 181 Kolmogorov-Smirnov is misspelled
L218, 'nests' not 'nest'
L221 delete 'as'
Figure 4 Remove 'a' from the ordinate label [should be 'Time of Day (h)']